# Self-sperm induce resistance to the detrimental effects of sexual encounters with males in hermaphroditic nematodes

Lauren N Booth[1], Travis J Maures[1†], Robin W Yeo[1], Cindy Tantilert[1], Anne Brunet[1,2]*

[1]Department of Genetics, Stanford University, Stanford, United States; [2]Glenn Laboratories for the Biology of Aging at Stanford University, Stanford, United States

**Abstract** Sexual interactions have a potent influence on health in several species, including mammals. Previous work in *C. elegans* identified strategies used by males to accelerate the demise of the opposite sex (hermaphrodites). But whether hermaphrodites evolved counter-strategies against males remains unknown. Here we discover that young *C. elegans* hermaphrodites are remarkably resistant to brief sexual encounters with males, whereas older hermaphrodites succumb prematurely. Surprisingly, it is not their youthfulness that protects young hermaphrodites, but the fact that they have self-sperm. The beneficial effect of self-sperm is mediated by a sperm-sensing pathway acting on the soma rather than by fertilization. Activation of this pathway in females triggers protection from the negative impact of males. Interestingly, the role of self-sperm in protecting against the detrimental effects of males evolved independently in hermaphroditic nematodes. Endogenous strategies to delay the negative effect of mating may represent a key evolutionary innovation to maximize reproductive success.

DOI: https://doi.org/10.7554/eLife.46418.001

*For correspondence:
abrunet1@stanford.edu

Present address: †Synthego, Redwood City, United States

## Introduction

Animals interact with each other in complex ways that can affect their health, including sexual, cooperative, and competitive interactions. Sexual interactions drastically impact an individual's health and behavior. In *Drosophila* and *C. elegans*, sexual interactions are detrimental to health and shorten the lifespan of both sexes (*Aprison and Ruvinsky, 2016*; *Chapman et al., 1995*; *Fowler and Partridge, 1989*; *Gems and Riddle, 1996*; *Gendron et al., 2014*; *Harvanek et al., 2017*; *Maures et al., 2014*; *Partridge and Farquhar, 1981*; *Promislow, 2003*; *Promislow, 1992*; *Shi and Murphy, 2014*; *Shi et al., 2017*; *Ting et al., 2014*; *Van Voorhies, 1992*; *Woodruff et al., 2014*; *Wu et al., 2012*). In mammals, mates induce neurological, developmental, and behavioral changes in the opposite sex and can negatively impact health (*Gao et al., 2017*; *Garratt et al., 2016*; *Aloise King et al., 2013*; *Stowers and Kuo, 2015*). For example, the presence of males can increase female body weight (*Garratt et al., 2016*) and accelerate puberty (*Flanagan et al., 2011*; *Vandenbergh, 1969*). Identification of specific strategies involved in responding to the effects of sexual encounters could help to improve our understanding of how the sexes interact with each other, how sexual pressures have shaped species over evolutionary time, and how these mechanisms could be harnessed to improve health.

*C. elegans* is particularly well suited to the study of the effect of sexual interactions on lifespan. Males shorten the lifespan of the *C. elegans* hermaphrodite through a phenomenon called male-induced demise (*Gems and Riddle, 1996*; *Maures et al., 2014*; *Shi and Murphy, 2014*). Previous studies showed that males shorten hermaphrodite lifespan by several means (*e.g.* sperm, seminal

**eLife digest** A nematode worm known as *Caenorhabditis elegans* is often used in the laboratory to study how animals grow and develop. There are two types of *C. elegans* worm: hermaphrodite individuals produce both female sex cells (eggs) and male sex cells (sperm), while male individuals only produce sperm.

The hermaphrodite worms are able to reproduce without mating with another worm, allowing populations of *C. elegans* to grow rapidly when they are living in favorable conditions. However, when the hermaphrodites do mate with males they tend to produce more offspring. These offspring are also usually healthier because they receive a mixture of genetic material from two different parents.

Although mating is beneficial for the survival of a species it can also harm an individual animal. Previous studies have shown that mating with male worms can accelerate aging of hermaphrodite worms and cause premature death. However, it remained unclear whether hermaphrodite worms have evolved any mechanisms to protect themselves after mating with a male.

To address this question, Booth et al. used genetic techniques to study the lifespans of hermaphrodite worms. The experiments found that the hermaphrodites' own sperm (known as self-sperm) regulated a sperm-sensing signaling pathway that protected them from the negative impact of mating with males. Hermaphrodites with self-sperm that mated with males lived for a similar length of time as hermaphrodites that did not mate. On the other hand, hermaphrodites that did not have self-sperm (because they were older or had a genetic mutation) had shorter lifespans after mating than worms that did not mate. Modulating the sperm-sensing signaling pathway in worms that lacked self-sperm was sufficient to protect them from the negative effects of mating with males.

Further experiments found that the hermaphrodites of another nematode worm called *C. briggsae* – which evolved self-sperm independently of *C. elegans* – also protected themselves from the negative effects of mating with males in a similar way. This suggests that other animals may also have evolved similar mechanisms to protect themselves from harm when mating.

A separate study by Shi et al. has found that the beneficial effects of self-sperm are mediated by a pathway linked to longevity that also exists in mammals. The results of both investigations combined suggest possible avenues for future research into the complex relationship between health, longevity, and reproduction.

DOI: https://doi.org/10.7554/eLife.46418.002

fluid, and pheromones), and identified some of the hermaphrodite genes that mediate male-induced demise (e.g. *utx-1*, *ins-11*, *daf-16*) (*Maures et al., 2014*; *Shi and Murphy, 2014*). However, whether hermaphrodites evolved natural defense mechanisms to protect themselves after a sexual encounter with a male remained unknown.

Here we discover that young hermaphrodites are entirely protected from brief sexual encounters with males. Surprisingly, the natural protection in young hermaphrodites is not due to their youthfulness, but rather to the presence of self-sperm. Self-sperm act through a sperm-sensing pathway to protect the soma in a fertilization-independent manner, and activating this pathway in females can protect them against the detrimental effect of males. The protective effect of self-sperm is conserved in other hermaphroditic nematodes and may represent a key adaptation for their reproductive success.

## Results

### Young hermaphrodites are protected from demise induced by a brief mating with males

Previous studies were done with long encounters between young *C. elegans* hermaphrodites and males (*Gems and Riddle, 1996*; *Maures et al., 2014*; *Shi and Murphy, 2014*), which does not reflect the situation in nature where males are rare (*Barrière and Félix, 2005*). We thus asked if varying the length of sexual interactions, as well as the age of males and hermaphrodites, could reveal natural strategies that evolved to mitigate the negative impact of sexual encounters on health

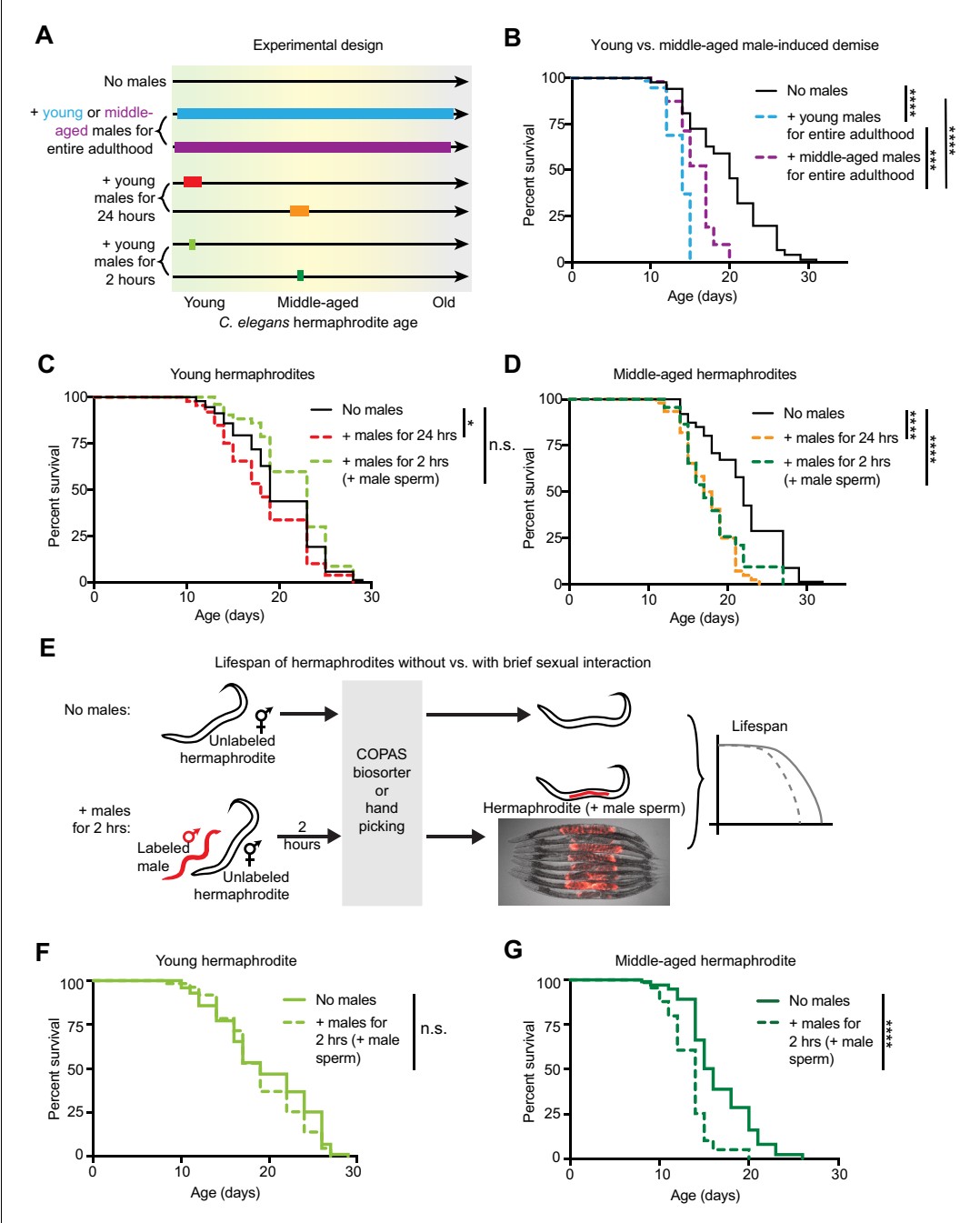

**Figure 1.** The length of sexual encounters and age of the sexual partners influences the detrimental effect of males on hermaphrodite lifespan. (A) Scheme describing the lengths of sexual interactions between *C. elegans* males and hermaphrodites and the ages of the sexual partners used in this study. Young was defined as the first day of adulthood (day 3 of life), and middle-aged as the 5th day of adulthood (day 7 of life). (B) Long and frequent sexual interactions with young males (blue dashed line) reduced hermaphrodite lifespan (p<0.0001 vs. no males). Long and frequent sexual interactions with middle-aged males (purple dashed line) also reduced hermaphrodite lifespan compared to hermaphrodites without males (p<0.0001), but middle-aged males shortened hermaphrodite lifespan less than young males (p=0.0003). (C-D) Hermaphrodites that interacted with males for 24 hr when young from day 3 to 4 of life (C, red dashed line) or middle-aged from day 6 to 7 of life (D, orange dashed line) lived shorter than hermaphrodites that never interact with males (p=0.029 and p<0.0001, respectively). Hermaphrodites that mated within a brief (2 hr) interaction with males when young on day 3 of life (C, lime green dashed line) did not have a shortened lifespan (n.s. vs. no males) but hermaphrodites that mated within a brief period (2 hr) when middle-aged on day 7 of life (D, green dashed line) did have a significantly shortened lifespan (p<0.0001 vs. no males). (E) To control for the mating efficiency differences between ages and genotypes (see *Supplementary file 3*), we measured the lifespans of only *C. elegans* individuals that had successfully mated by identifying and isolating hermaphrodites that have fluorescent male sperm (represented by the red tilde in the scheme) from

*Figure 1 continued on next page*

*Figure 1 continued*

those that are unmated and lack fluorescent male sperm using either hand picking or a large particle COPAS large particle biosorter. The presence of fluorescent male sperm is indicative of fertilization (*Figure 1—figure supplement 1B, C*), though this was not specifically measured for the lifespan assays. Hermaphrodites that received male sperm following two hours with males were compared to hermaphrodites that never interacted with males but that were hand-picked or run through the COPAS large particle biosorter. (**F-G**) Using a different method of isolating mated hermaphrodites, we also found that young hermaphrodites (**F**) were resistant to mating-induced demise if they received male sperm and seminal fluid during a brief, 2 hr interaction with males (n.s. vs. no males) but that older hermaphrodites (**G**) were sensitive and lived shorter following a brief, 2 hr interaction with males (p<0.0001 vs. no males). In panels C and D, hermaphrodites that received fluorescent male sperm were isolated by hand and the males were *him-5* (*e1467*) mutants. In panels F and G, males were *him-8*(*e1489*) mutants with a male-specific GFP reporter (*Ppkd-2::GFP*) and hermaphrodites with fluorescent male sperm were isolated with the COPAS large particle biosorter. For each condition, 67–114 animals were used to quantify lifespan. Lifespan data are plotted as Kaplan-Meier survival curves and *p*-values were determined using Mantel-Cox log ranking. *p<0.05, **p<0.01, ***p<0.001, ****p<0.0001, n.s. = not significant. See also *Supplementary file 2* for extended statistics and replicates.

DOI: https://doi.org/10.7554/eLife.46418.003

The following figure supplement is available for figure 1:

**Figure supplement 1.** Effect of long-term exposure to young males and development of the sperm tracking method.

DOI: https://doi.org/10.7554/eLife.46418.004

(*Figure 1A*). As previously shown (*Gems and Riddle, 1996*; *Maures et al., 2014*; *Shi and Murphy, 2014*), long and frequent sexual interactions with young males shortened hermaphrodite lifespan (*Figure 1B* and *Figure 1—figure supplement 1A*). Middle-aged males were less able to induce premature demise of the opposite sex than young males (*Figure 1B*), probably because of their decreased ability to efficiently mate (*Guo et al., 2012*) (*Supplementary file 3*).

To better mimic the natural situation—where sexual interactions are infrequent due to the rarity of males—and isolate the mating-specific aspect of sexual interactions, we tested how a brief (two-hour) mating with young males impacts young and middle-aged hermaphrodites (*Figure 1C,D*). Because the chance of a sexual interaction is relatively low in the brief two-hour exposure and can differ depending on the age of the hermaphrodite (*Supplementary file 3; Garcia et al., 2007*; *Leighton et al., 2014*), we specifically measured the lifespans of hermaphrodites that mated using a fluorescent sperm tracking method to isolate hermaphrodites that received male sperm (*Figure 1E* and *Figure 1—figure supplement 1B–F*) (*Stanfield and Villeneuve, 2006*). Interestingly, while a brief encounter with males induced the premature death of middle-aged hermaphrodites, it did not affect the lifespan of young hermaphrodites (*Figure 1C,D,F and G*). Young hermaphrodites were remarkably resistant to the negative impact of males, even though they successfully mated (*Figure 1C,F*). We confirmed this observation using two different male strains and two methods of isolating mated hermaphrodites (*Figure 1C,D,F and G*). Thus, shortening the length of sexual interactions, which is more similar to the situation in nature, reveals that young hermaphrodites can defend themselves against the lifespan-shortening effect of mating with males.

## The presence of self-sperm, not youthfulness, is necessary for the protection of hermaphrodites from the negative impact of mating with males

Young and middle-aged hermaphrodites differ not only by their ages, but also in their reproductive status. Young hermaphrodites are self-fertile, due to the presence of both oocytes and self-sperm. In contrast, middle-aged hermaphrodites are no longer self-fertile, because they have exhausted their self-sperm, though they can still reproduce if their oocytes are fertilized by males (*Riddle et al., 1997*). To disentangle age and reproductive status, we compared the lifespan of young hermaphrodites to that of young 'feminized' individuals (which have oocytes, but no self-sperm; *Ellis and Schedl, 2007*) after a brief encounter with males. Surprisingly, young feminized individuals (*fem-1* [*hc17*] or *fog-2*[*q71*]; *Doniach and Hodgkin, 1984*; *Schedl and Kimble, 1988*) were sensitive to mating-induced death and exhibited premature death and deterioration (*Figure 2A–D* and *Figure 2—figure supplement 1A,B*). Although mating efficiency can vary between *C. elegans* mutants and individuals of different ages (*Supplementary file 3*; *Garcia et al., 2007*; *Leighton et al., 2014*; *Morsci et al., 2011*), we specifically measured the lifespan of only the hermaphrodites that mated (i.e. received male sperm) (*Figure 1E*). Thus, mating efficiency differences are unlikely to contribute to the sensitivity of feminized individuals to males. Consistent with the observation that the presence

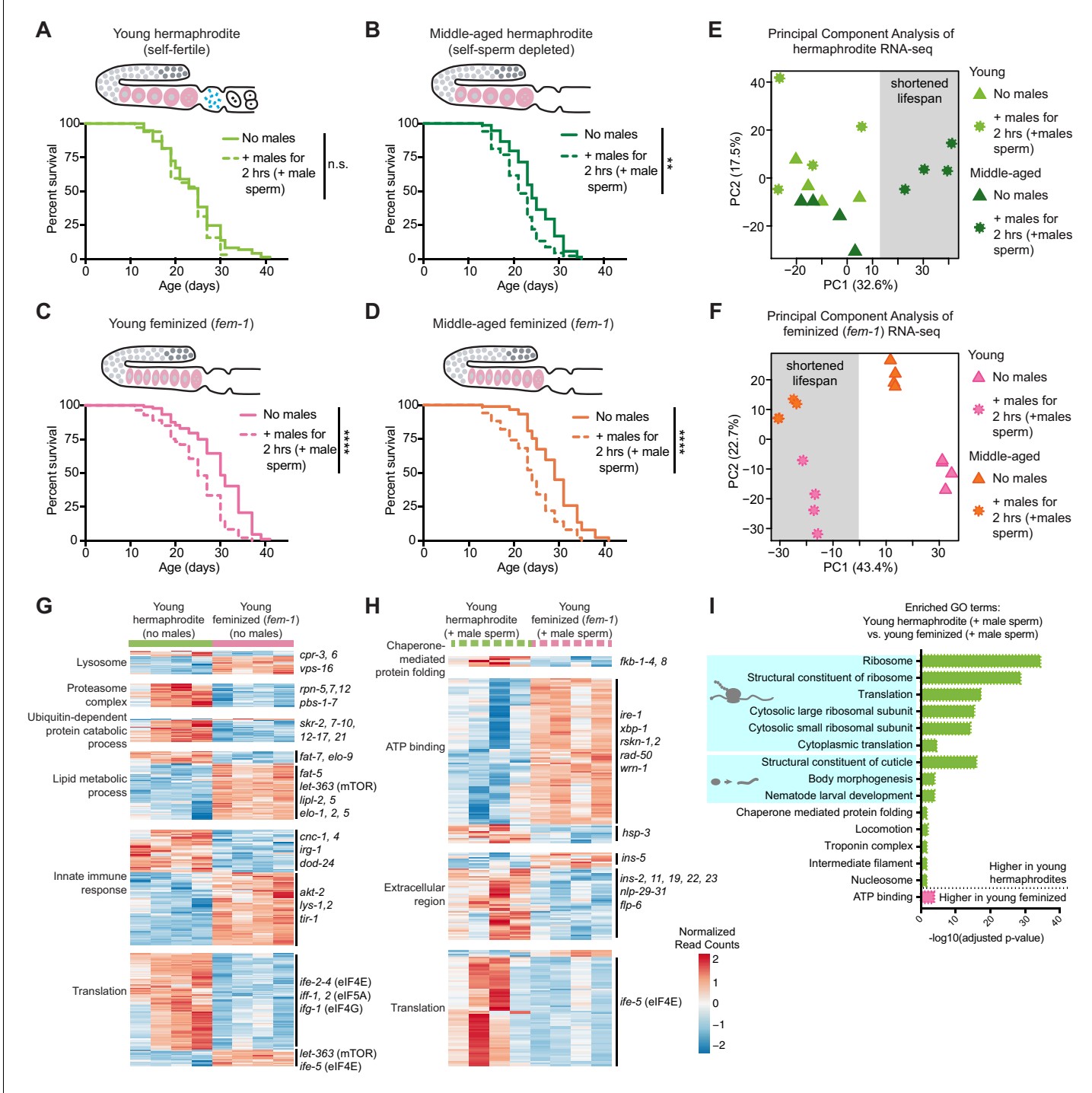

**Figure 2.** The presence of self-sperm is necessary for the resistance of young hermaphrodites to a brief encounter with males. (A-B) Young, self-fertile, wild-type hermaphrodites with self-sperm (A, day 3 of life) that received male sperm after a brief interaction with males had a normal lifespan (n.s. vs. no males) whereas middle-aged, wild-type hermaphrodites that are self-sperm depleted (B, day 7 of life) had a shortened lifespan (p=0.0013 vs. no males). (C-D) The lifespan of feminized *C. elegans* that lack self-sperm at all ages (*fem-1[hc17]*) was reduced if they received male sperm during a brief, 2 hr interaction with males either when young (C, day 3 of life, p<0.0001 vs. no males) or middle-aged (D, day 7 of life, p<0.0001 vs. no males). Images above the lifespan curves in panels A-D show the state of the germline with oocytes in pink and self-sperm in blue. (E-F) Principal Component Analysis (PCA) of the normalized read counts from the entire transcriptomes of hermaphrodite (E) and feminized (F) *C. elegans* that never interacted with males and that received male sperm during a two-hour interaction with males. (G-H) Heatmaps of the normalized read counts for the differentially expressed genes that comprise select GO terms that were enriched when comparing young hermaphrodite and young feminized (*fem-1*) individuals without male

*Figure 2 continued on next page*

*Figure 2 continued*

exposure (G) and that received male sperm following a brief interaction with males (H). The four replicates for each condition are shown. (I) Selected, enriched GO terms from the differentially expressed genes between young hermaphrodite versus young feminized that received male sperm. GO terms that were enriched in the genes expressed more highly in young hermaphrodites are shown in green and GO terms enriched in the genes more highly expressed in young feminized individuals are in pink. *P*-values were calculated with the Fisher's exact test and corrected for multiple hypothesis testing with Benjamini-Hochberg. A complete list of all significantly enriched GO terms can be found in *Figure 2—source data 2*. All individuals were raised at the restrictive temperature (25°C) until the onset of adulthood (day 3 of life) and then moved to 20°C for the remainder of their lifespan. Worms that received male sperm from *him-5(e1467)* males were isolated by hand picking individuals with fluorescent male sperm in their uterus or spermatheca. For all lifespan assays 52–109 animals were tested per condition. Lifespan data are plotted as Kaplan-Meier survival curves and *p*-values were determined using Mantel-Cox log ranking. *p<0.05, **p<0.01, ***p<0.001, ****p<0.0001, n.s. = not significant. See also *Supplementary file 2* for extended statistics and replicates.

DOI: https://doi.org/10.7554/eLife.46418.005

The following source data and figure supplements are available for figure 2:

**Source data 1.** The DESeq2 output (differential expression) from the RNA-seq analysis.

DOI: https://doi.org/10.7554/eLife.46418.008

**Source data 2.** The complete list of GO terms whose enrichment was determined using the significantly differentially expressed genes when comparing young hermaphrodites vs. young feminized individuals (selected, enriched GO results are displayed in *Figure 2G–I* and in *Figure 2—figure supplement 2C*).

DOI: https://doi.org/10.7554/eLife.46418.009

**Figure supplement 1.** The effect of a brief encounter with males on feminized and sterile individuals.

DOI: https://doi.org/10.7554/eLife.46418.006

**Figure supplement 2.** RNA-seq of young and middle-aged hermaphrodites and feminized individuals with and without receiving male sperm.

DOI: https://doi.org/10.7554/eLife.46418.007

of self-sperm is necessary to live a normal lifespan even after a brief encounter with males, young individuals that lack self-sperm and oocytes (*glp-1[e2144]*; *Priess et al., 1987*) also lived shorter after a brief encounter with males (*Figure 2—figure supplement 1C,D*), even though *glp-1* mutants are normally long-lived (*Antebi, 2013*; *Arantes-Oliveira et al., 2002*; *Hansen et al., 2013*; *Keith and Ghazi, 2015*; *Wang et al., 2008*). Together, these results indicate that a major factor in protecting an individual from the detrimental influence of males is not youthfulness, but the presence of self-sperm.

To further investigate the importance of self-sperm in the resistance of young hermaphrodites to the negative impact of mating with males, we performed RNA-seq on young or middle-aged hermaphrodites or feminized individuals that were never exposed to males (no males) or had a brief encounter with males (+male sperm) (*Figure 2—figure supplement 2A*). Principal Component Analysis (PCA) revealed that, as expected (*Angeles-Albores et al., 2017*), the transcriptomes of hermaphrodites and feminized individuals that were never exposed to males separated based on fertility status and, to some extent, age (*Figure 2E,F* and *Figure 2—figure supplement 2B*). Interestingly, the first principal component separated the transcriptomes of hermaphrodites that have a shortened lifespan (middle-aged upon mating) from those of hermaphrodites that live a normal lifespan (young upon mating, young and middle-aged without mating) (*Figure 2E*). Consistently, the first principal component also separated the transcriptomes of feminized individuals that have a shortened lifespan (young and middle-aged upon mating) from those of feminized individuals that live a normal lifespan (young and middle-aged without mating) (*Figure 2F*). This observation confirms the importance of self-sperm in the protection from males. GO terms linked with individuals that are resistant (e.g. young hermaphrodites) or sensitive to mating-induced death (e.g. young feminized) included translation, lipid metabolism, the innate immune response, and protein homeostasis (proteostasis) (*Figure 2G–I*, *Figure 2—figure supplement 2C*, and *Figure 2—source data 1* and *2*), suggesting these conserved homeostatic pathways could be responsible for the lifespan differences in the response to males. Indeed, many of these pathways are linked with longevity (*Kaeberlein et al., 2015*; *Kenyon, 2010*; *López-Otín et al., 2013*; *Riera et al., 2016*; *Shore and Ruvkun, 2013*). Collectively, these data indicate that the presence of self-sperm is necessary to protect the soma against deterioration due to brief encounters with males, perhaps by regulating homeostasis pathways.

## The presence of self-sperm is sufficient to protect from the detrimental effects on lifespan of a brief mating with males

Self-sperm play a key role in germline quality assurance in *C. elegans* by triggering the clearance of carbonylated and aggregated proteins from the germline (*Bohnert and Kenyon, 2017*; *Goudeau and Aguilaniu, 2010*). However, whether self-sperm could harness this potential to protect the soma is not known. To determine if the presence of self-sperm is sufficient to protect hermaphrodites, we examined the response of individuals with a masculinized germline (which have a female soma and self-sperm, but no oocytes), *fem-3(q20)* (*Ahringer and Kimble, 1991*; *Barton et al., 1987*; *Ellis and Schedl, 2007*). Masculinized individuals that only have self-sperm were protected from premature death induced by mating with males not only when young, but even at an older age when hermaphrodites normally become sensitive to an encounter with males (*Figure 3A–D*, and *Figure 3—figure supplement 1A,B*). These data suggest that the presence of self-sperm is sufficient to protect from the detrimental effects on lifespan of a brief mating with males.

We next asked whether self-sperm could act via their ability to self-fertilize hermaphrodites. Interestingly, hermaphrodite mutants that are defective in self-fertilization despite maintaining self-sperm (*spe-9[hc88]*; *Ellis and Stanfield, 2014b*; *Singson et al., 1998*) were still protected from early death induced by mating with males when young (*Figure 3E,F*). In contrast, hermaphrodites that are defective in self-sperm maturation (due to loss of the SPE-44 transcription factor; *Kasimatis et al., 2018*; *Kulkarni et al., 2012*) were no longer protected from the lifespan shortening effects of mating with males when young (*Figure 3G* and *Figure 3—figure supplement 1C,D*). Thus, the presence of mature self-sperm, but not self-fertilization, is sufficient for the protection against mating-induced demise.

## Self-sperm protect hermaphrodites by triggering a sperm-sensing pathway that normally affects the germline to protect the soma

We next explored the mechanisms by which self-sperm protect individuals. Self-sperm could protect individuals independently of fertilization, by acting via sperm-sensing pathways. In *C. elegans*, sperm proteins are known to be sensed by the somatic gonad and to signal to the germline and somatic gonad by repressing the homeodomain transcription factor CEH-18 and the Ephrin receptor VAB-1 (*Figure 4A* and *Greenstein et al., 1994*; *Miller, 2001*; *Miller, 2003*). Hence, the loss of CEH-18 and VAB-1 can mimic the presence of self-sperm (*Miller, 2003*). Deficiency in CEH-18 and VAB-1 was sufficient to protect middle-aged hermaphrodites that have depleted their self-sperm from the negative effects of mating with males (*Figure 4D,E*). Interestingly, deficiency in CEH-18 and VAB-1 in young feminized individuals made them resistant to brief encounter with males (*Figure 4B,C*), indicating that this conserved pathway could be sufficient to protect females against the negative impact of males.

To understand how this pathway might work to protect the soma, we examined the potential links between the transcription factor CEH-18 and the genes differentially expressed between hermaphrodites and feminized individuals that we identified by RNA-seq. *ceh-18* mRNA levels were not significantly affected by the presence of self-sperm or mating with a male (*Figure 4—figure supplement 1A*). In contrast, several genes that interact genetically or physically with CEH-18, including the chromatin modifiers *set-33* and *jmjd-3.2*, were differentially expressed between young hermaphrodites and feminized individuals that have successfully mated with males (*Figure 4F* and *Figure 4—figure supplement 1B*). These observations suggest that differences in chromatin state between young hermaphrodites and feminized individuals may contribute to the transcriptional effect of the CEH-18 sperm-sensing transcription factor in the resistance and sensitivity to mating-induced death. A subset of the genes regulated by the presence of self-sperm or mating with a male contained a CEH-18 binding site in their regulatory regions (*Figure 4—source data 1*), though the CEH-18 motif and binding peaks (*Kudron et al., 2018*; *Narasimhan et al., 2015*) were not significantly enriched in these genes (perhaps due to cell heterogeneity (*Cao et al., 2017*), see Material and methods). We asked if the subset of genes that are differentially expressed between young hermaphrodites and young feminized individuals and contain a CEH-18 binding site were enriched for specific biological features (*Figure 4—source data 1* and *2*). Interestingly, GO term enrichment of these genes revealed several terms linked with longevity, including 'Determination of adult lifespan'

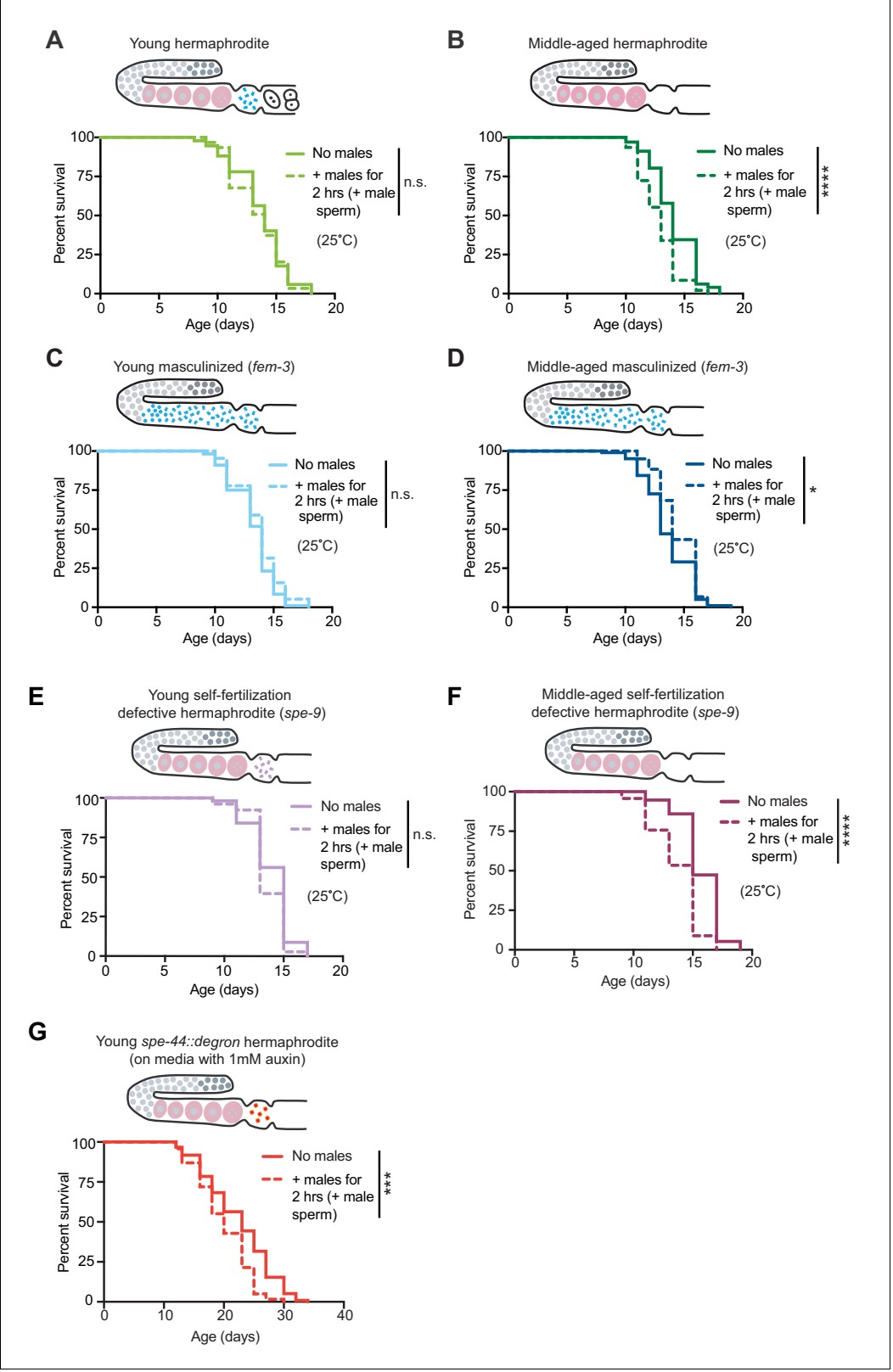

**Figure 3.** The presence of self-sperm is sufficient for the resistance to mating-induced demise. (A-D) Hermaphrodites with a masculinized germline (*fem-3[q20]*) that have self-sperm when young (C, day 3 of life) and middle-aged (D, day 6 of life), were resistant to a brief, 2 hr interaction with males and lived a normal lifespan. As a control, wild-type hermaphrodites (A and B) were also tested and showed the expected resistance to a brief

*Figure 3 continued on next page*

*Figure 3 continued*

interaction with males when young and shortened lifespan following a brief interaction with males when middle-aged. (E-F) Young hermaphrodites that have fertilization-defective self-sperm (*spe-9[hc88]*, panel E, day 3 of life) were protected from a brief, 2 hr interaction with males when young (n.s. vs. no males) but when self-sperm are depleted with age (F, day 7 of life), their lifespan was shortened if they received male sperm during brief interaction with males (p<0.0001 vs. no males). (G) Young hermaphrodites that have defective sperm due to the absence of the SPE-44 transcription factor (*fxIs1[pie-1p::TIR1::mRuby]; spe-44(fx110[spe-44::degron])* grown on 1 mM auxin until adulthood), are not fully protected from a brief, 2 hr interaction with males (p=0.0001 vs. no males). Images above the lifespan curves show the state of the germline. Mated worms (dashed lines) were selected by hand-picking based on the presence of fluorescent male sperm in their uterus or spermatheca following a two-hour interaction with *him-5(e1467)* males. Lifespans were performed with 31–144 animals per condition. Masculinized (*fem-3[q20]*) versus WT and fertilization-defective self-sperm (*spe-9[hc88]*) experiments (A-F) were performed at the restrictive temperature, 25°C. The SPE-44 auxin-inducible degradation experiment (G) was performed at 20°C and controls for this experiment are found in *Supplementary file 2* and *Figure 3—figure supplement 1C–D*. Lifespan data are plotted as Kaplan-Meier survival curves and *p*-values were determined using Mantel-Cox log ranking. *p<0.05, **p<0.01, ***p<0.001, ****p<0.0001, n.s. = not significant. See also *Supplementary file 2* for extended statistics and replicates.

DOI: https://doi.org/10.7554/eLife.46418.010

The following figure supplement is available for figure 3:

**Figure supplement 1.** The effect of males on hermaphrodites and masculinized individuals.

DOI: https://doi.org/10.7554/eLife.46418.011

---

(*Figure 4G*, *Figure 4—figure supplement 1C*), suggesting that the sperm-sensing transcription factor CEH-18 may directly regulate genes that affect longevity.

Together, these results show that self-sperm protect individuals, even older ones, from mating-induced death by repressing a sperm-sensing pathway in the somatic gonad, which could in turn protect hermaphrodites by altering chromatin and transcriptional networks.

## The ability of self-sperm to promote resistance to mating with males evolved independently twice in nematodes

Is the ability of self-sperm to promote resistance to brief interactions with males unique to *C. elegans* or a common hermaphroditic strategy for resistance to the negative impact of mating with males to optimize their reproduction and health? The *C. elegans* ancestor was a gonochoric species with true females and obligatory males (*Kiontke et al., 2004*). Hermaphroditism evolved at least three times independently in this lineage (in *C. elegans*, *C. briggsae*, and *C. tropicalis*) (*Ellis and Lin, 2014a*; *Guo et al., 2009*; *Hill et al., 2006*; *Kiontke et al., 2004*; *Kiontke et al., 2011*; *Nayak et al., 2005*; *Thomas et al., 2012*) (*Figure 5A*). Other nematode species remained true females (*e.g. C. remanei* and *C. brenneri*) (*Figure 5A*).

Both hermaphroditic and true female nematodes are known to succumb prematurely following long interactions with males (*Figure 4—figure supplement 1A,B* and *Maures et al., 2014*; *Palopoli et al., 2015*; *Shi and Murphy, 2014*), but the effect of brief interactions has not been tested. We found that *C. remanei* and *C. brenneri* females lived normal lifespans following brief, two-hour interactions with males at all of the ages we tested (*Figure 5B* and *Figure 5—figure supplement 1B*), suggesting that the differential sensitivity of old versus young hermaphrodites may have evolved in hermaphroditic species. Indeed, we found that young *C. briggsae* hermaphrodites (which have self-sperm) resisted a brief exposure to males and lived a normal lifespan whereas middle-aged *C. briggsae* hermaphrodites (which have depleted their self-sperm) succumbed to brief encounter with males (*Figure 5C* and *Figure 5—figure supplement 1C*). Interestingly, feminized individuals in *C. briggsae* (due to a different mutation than *C. elegans*, the *she-1* feminizing mutation, which affects sex determination; *Guo et al., 2009*), also became sensitive to a brief interaction with males when young and exhibited premature death (*Figure 5D*). As with *C. elegans*, we controlled for age- and genotype-dependent differences in *C. briggsae* mating efficiency (*Supplementary file 3*) using fluorescent male sperm tracking (*Figure 1E*).

Does the sperm-sensing pathway mediate self-sperm protection from mating-induced demise in *C. briggsae* as it does in *C. elegans* hermaphrodites? We first verified that knock-down of the orthologous sperm-sensing pathway in *C. briggsae* females (*mfIs42[Cel-sid-2; Cel-myo-2::DsRed]; she-1*

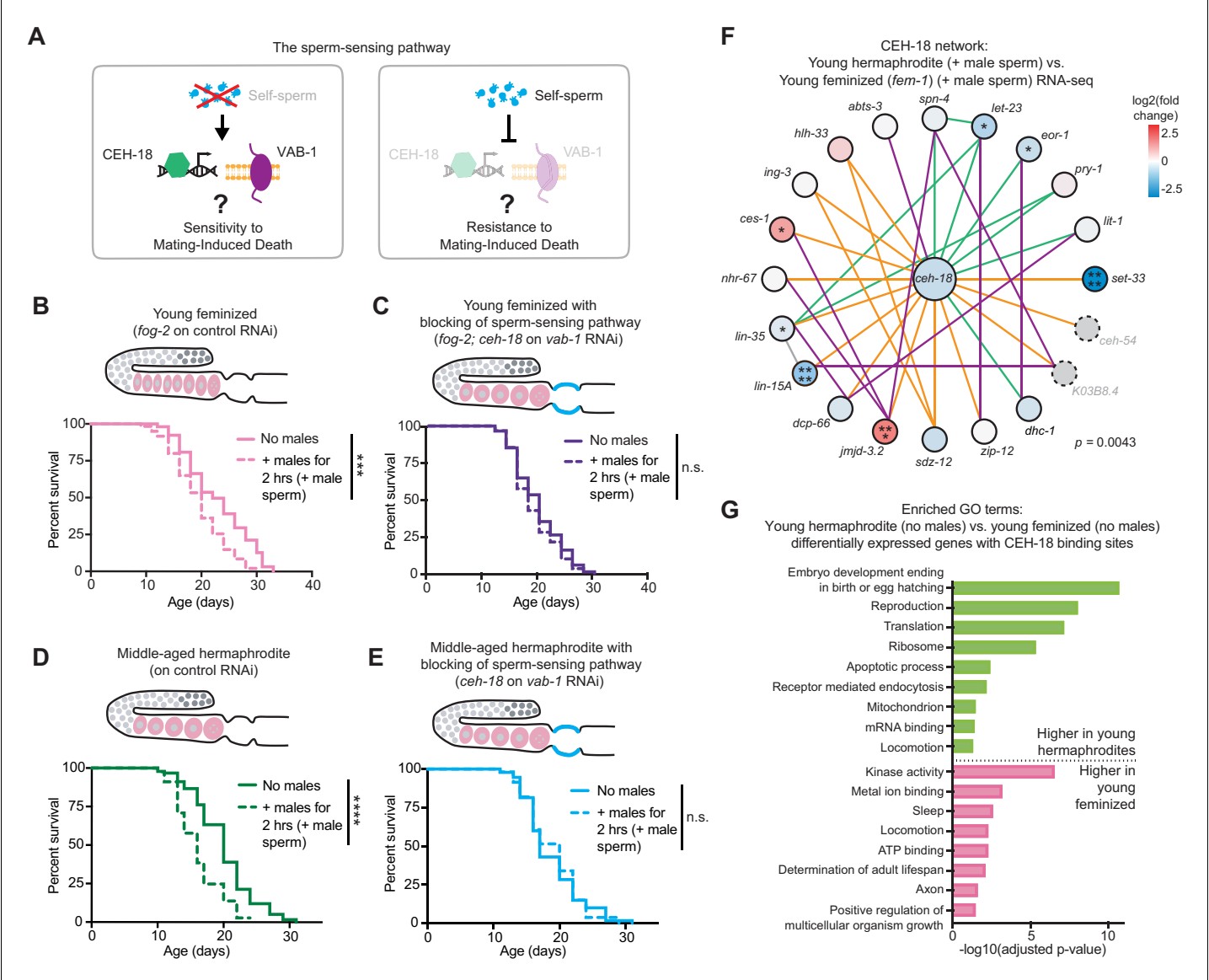

**Figure 4.** Self-sperm act via repression of a somatic sperm-sensing pathway to mediate resistance to mating-induced demise. (**A**) A model for the role of self-sperm in mating-induced demise resistance. The absence of self-sperm (right panel) activates the CEH-18 and VAB-1 sensing pathway in the somatic gonad. (**B-C**) Young, feminized (*fog-2[q71]*) worms (**B**, day 4 of life) were sensitive to a brief interaction with males when young (B, p=0.0001 vs. no males) but when the sperm-sensing pathway was blocked by loss of CEH-18 and VAB-1 (*fog-2[q71]; ceh-18[mg57]* grown on *vab-1* targeting RNAi bacteria), these worms (**C**) were resistant to a brief, 2 hr interaction with males and had a normal lifespan (n.s. vs. no males). (**D-E**) Middle-aged, self-sperm depleted hermaphrodites (**D**, day 7 of life) were sensitive to a brief mating with males (D, p<0.0001 vs. no males) but when the sperm-sensing pathway was blocked by loss of CEH-18 and VAB-1 (*ceh-18[mg57]* grown on *vab-1* targeting RNAi bacteria), these middle-aged worms (**E**) were resistant to a brief, 2 hr interaction with males and had a normal lifespan (E, n.s. vs. no males). For all experiments, worms that received male sperm (dashed lines) were selected by hand-picking based on the presence of fluorescent male sperm in their uterus or spermatheca following a two-hour interaction with *him-5(e1467)* males. Images above the lifespan curves in panels B-E show the state of the germline with oocytes in pink and the blocking of sperm-sensing pathway by a blue line. Lifespans were performed with 86–115 animals per condition. Lifespan assays were performed at 20° C. Lifespan data are plotted as Kaplan-Meier survival curves and *p*-values were determined using Mantel-Cox log ranking. *p<0.05, **p<0.01, ***p<0.001, ****p<0.0001, n.s. = not significant. See also ***Supplementary file 2*** for extended statistics and replicates. (**F**) The CEH-18 interaction network calculated by GeneMANIA (***Warde-Farley et al., 2010***). Lines indicate genetic interactions (green), co-expression (purple), physical interactions (orange), and 'other' (gray). The differential expression of the genes in the network is shown with a blue-red gradient. *p<0.05, **p<0.01, ***p<0.001, ****p<0.0001, no stars = not significant. Network members not detected by the RNA-seq are in gray with dotted outlines. The number of differentially expressed genes in this network was statistically enriched (p=0.0043) as measured using the hypergeometric distribution test. See ***Figure 2—source data 1*** for exact differential expression values. (**G**) Selected, enriched GO terms from the genes that are differentially expressed between young hermaphrodite versus young feminized that never interacted with a male and contain a CEH-18 binding site as defined by CEH-18 ChIP-seq

*Figure 4 continued*

(*Kudron et al., 2018*), see Material and methods. GO terms that were enriched in the genes expressed more highly in young hermaphrodites are shown in green and GO terms enriched in the genes more highly expressed in young feminized individuals are in pink. *P*-values were calculated with the Fisher's exact test and were corrected for multiple hypothesis testing with Benjamini-Hochberg. A complete list of all significantly enriched GO terms can be found in *Figure 4—source data 2*.

DOI: https://doi.org/10.7554/eLife.46418.012

The following source data and figure supplement are available for figure 4:

**Source data 1.** The intersection of the DESeq2 output (differential expression) and the CEH-18 binding sites (*Kudron et al., 2018*).

DOI: https://doi.org/10.7554/eLife.46418.014

**Source data 2.** The complete list of GO terms whose enrichment was determined using the significantly differentially expressed genes associated with CEH-18 binding peaks when comparing young hermaphrodites vs. young feminized individuals (selected, enriched GO results are displayed in *Figure 4G* and *Figure 4—figure supplement 1C*).

DOI: https://doi.org/10.7554/eLife.46418.015

**Figure supplement 1.** CEH-18 expression and network analysis.

DOI: https://doi.org/10.7554/eLife.46418.013

*[v35]*) indeed reduced the accumulation of unfertilized oocytes in the gonad arms and increased the number of unfertilized oocytes in the uterus (*Figure 5—figure supplement 3*) similar to *C. elegans* (*Miller, 2003*). Interestingly, knock-down of the *C. briggsae* sperm-sensing genes protected middle-aged hermaphrodites (which are self-sperm depleted) from the lifespan shortening effects of a brief interaction with males (*Figure 5E,F*).

Together, these results suggest that hermaphroditism, notably the presence of self-sperm and detection by the sperm-sensing pathway, may have co-evolved more than once with strategies to protect from the detrimental effect of sexual interactions with the opposite sex. This may be particularly important to allow hermaphrodites to maximize their reproductive success by fertilizing their eggs through self-fertilization and mating with another male later in life. As male sperm out-compete hermaphroditic self-sperm (*LaMunyon and Samuel, 1999*; *Ward and Carrel, 1979*), such a strategy would also allow for the hermaphroditic self-sperm to be used. Because the sperm-sensing pathway is conserved in species with true females and is sufficient to protect feminized individuals, this pathway could be leveraged to protect the soma in a conserved manner, even in other species.

## Discussion

Here we show that a sensing mechanism between the germline and the soma mediates protection against the negative impact of sexual interactions. While previous work revealed the phenomenon of male-induced demise and identified specific mutations that could protect individuals from male-induced demise (*Gems and Riddle, 1996*; *Maures et al., 2014*; *Shi and Murphy, 2014*), it was unknown whether natural defenses exist to protect from the detrimental effects of males and what their mechanism of action could be. The previous studies investigated the effects of prolonged interactions between the sexes. While there is evidence of occasional 'bursts' of high rates of males in wild *C. elegans* populations (*Barrière and Félix, 2005*; *Frézal and Félix, 2015*; *Sivasundar and Hey, 2005*), males in the wild are thought to be rare. Thus, sexual interactions in nature are likely to be quite brief. Here, we have used very brief interactions between the sexes to uncover phenomena that better reflect the natural situation. We show for the first time that innate mechanisms that involve self-sperm can protect against the deleterious effects of mating with males.

The presence of self-sperm in hermaphrodites is also known to decrease attraction of and mating with males (*Garcia et al., 2007*; *Leighton et al., 2014*; *Morsci et al., 2011*; *Shi and Murphy, 2014*), and we did account for this in our study by selecting only mated individuals. Hence, self-sperm may be evolutionary important to ensure, in several different ways, that hermaphrodites produce as many of their own self-progeny as possible before succumbing to mating-induced death. Whether the ability of males to shorten the lifespan of older hermaphrodites that lack self-sperm is advantageous in nature remains unknown. The number of animals that reach middle or older-age in the wild is probably low. However, the protective effect of self-sperm on the soma appears to have evolved independently in two distantly-related species, suggesting that it is important for nematode hermaphrodites. The role of self-sperm as a signal to regulate the response to the opposite sex may

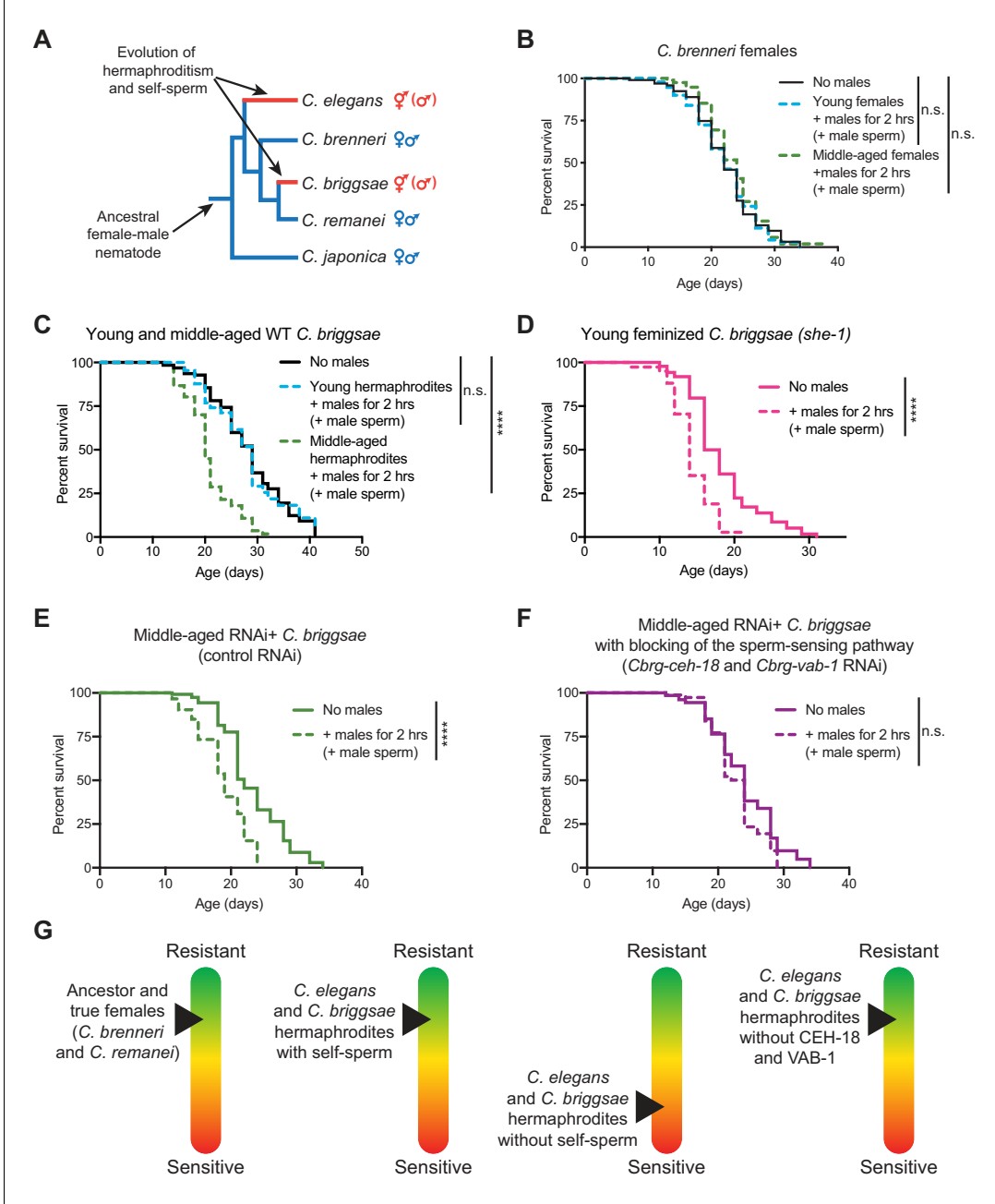

**Figure 5.** The importance of self-sperm in protecting young hermaphrodites against males independently evolved in a distantly-related nematode species. (A) The phylogeny of *Caenorhabditis* nematodes with the hermaphroditic lineages shown in red. (B) *C. brenneri* females lived a normal lifespan if they mated with a male during brief, two-hour interaction when young or middle-aged (n.s. vs. no males). For *C. brenneri*, we defined young as day 4 of life and middle-aged as day 10 of life. (C) *C. briggsae* hermaphrodites had a normal lifespan if they received male sperm during a brief interaction with males when they were young (day 3 of life, blue dashed line, n.s vs. no males) but had a shortened lifespan if they received male sperm when middle-aged (day 8 of life, green dashed line, p<0.0001 vs. no males). (D) Feminized *C. briggsae* (*she-1[v35]*) had a shortened lifespan if they received male sperm following a brief interaction with males when young (day 3 of life, p<0.0001 vs. no males). (E-F) Middle-aged, self-sperm depleted *C. briggsae* have a shortened lifespan following mating with a male (E, p<0.0001 vs. no males) but when the orthologs of the sperm-sensing pathway (*Cbr-ceh-18* and *Cbr-vab-1*) was blocked by RNAi knock-down, these worms lived a normal lifespan (F, n.s. vs. no males). Experiments were performed using a transgenic *C. briggsae* strain that is capable of RNAi knock-down by dsRNA ingestion (*mfls42[Cel-sid-2; Cel-myo-2::DsRed]*). (G) A scheme summarizing the resistance of sensitivity of different species, aged individuals, and mutants to mating-induced demise. Mated worms (dashed lines) were selected by hand-picking based on the presence of fluorescent male sperm in their uterus or spermatheca. *C. briggsae* males are *him-8(v186)* mutants (*Wei et al., 2014*). Lifespans were performed with 82–151 animals. Lifespan data are plotted as Kaplan-Meier survival curves and *p*-values were

*Figure 5 continued on next page*

Figure 5 continued

determined using Mantel-Cox log ranking. *p<0.05, **p<0.01, *p* < *** 0.001, ****p<0.0001, n.s. = not significant. See also *Supplementary file 2* for extended statistics and replicates.

DOI: https://doi.org/10.7554/eLife.46418.016

The following figure supplements are available for figure 5:

**Figure supplement 1.** The evolution of self-sperm mediated protection from mating-induced demise.

DOI: https://doi.org/10.7554/eLife.46418.017

**Figure supplement 2.** Conservation of CEH-18 and VAB-1.

DOI: https://doi.org/10.7554/eLife.46418.018

**Figure supplement 3.** RNAi knock-down of the sperm-sensing pathway in *C.briggsae*.

DOI: https://doi.org/10.7554/eLife.46418.019

also have evolved in other animals, including self-fertilizing hermaphroditic species (*e.g.* snails, slugs, and the vertebrate mangrove killifish) (*Jarne and Auld, 2006*; *Mackiewicz et al., 2006*).

The ancestor of *C. elegans* and related nematodes did not have self-sperm-mediated protection from mating-induced death and our data indicate that self-sperm mediated protection was an evolutionary innovation in the *C. elegans* and *C. briggsae* lineages. Whether the evolution of resistance to mating-induced death in *C. elegans* and *C. briggsae* occurred as the result of linking ancestral protection mechanisms to self-sperm, or whether novel mechanisms evolved in these hermaphroditic lineages, is unknown. However, given that self-sperm mediated mating-induced death protection evolved more than once in *Caenorhabditis* nematodes, the most parsimonious model is that the hermaphroditic lineages linked ancestral protection mechanisms with self-sperm. Indeed, the signaling pathways that are engaged by self-sperm (an Ephrin receptor and homeodomain transcription factor) are conserved in nematodes, suggesting that these ancestral pathways were harnessed in unique ways in the hermaphroditic lineages to evolve self-sperm mediated resistance to mating-induced death. This raises the intriguing possibility that such protective mechanisms could also be 'tuned up' in female species by mimicking activation of the signaling pathways that are triggered by self-sperm. Interestingly, the Ephrin receptor that detects the presence of sperm is conserved in mammals, it is expressed in the somatic gonad (*Figure 5—figure supplement 2B*; *Uhlén et al., 2015*), and is linked with mammalian fertility (*Barban et al., 2016*; *Buensuceso et al., 2016*). Thus, some of the mechanisms that signal between germline and soma could be used more generally to protect against the negative impact of sexual interactions in other species.

Fertilization has a rejuvenating effect in the germline and resets the aging clock in each generation (*Rando and Chang, 2012*; *Unal et al., 2011*). Consistently, self-sperm can clear carbonylated and aggregated proteins, two aging hallmarks, in the germline of *C. elegans* (*Bohnert and Kenyon, 2017*; *Goudeau and Aguilaniu, 2010*). Resetting of aging features can be achieved to some degree in mammalian somatic cells by the process of reprogramming to induced pluripotent stem cells, which can mimic fertilization and has a 'rejuvenating' capacity (*Rando and Chang, 2012*). In mammals, triggering rejuvenating factors in somatic cells, in a manner that is uncoupled from de-differentiation, has been recently suggested to be a potential rejuvenation strategy (*Ocampo et al., 2016*). However, the potential role of sperm in protection, and possible rejuvenation, of the soma is not known. Given the effect of sperm and fertilization on resetting the aging clock in the germline, as well as the conservation of key elements of the sensing pathway, it is possible that the protective properties of self-sperm, or even sperm itself, are conserved in other species, including mammals. These findings could open new strategies for harnessing the potential of gametes and their sensing pathways to 'reset' some hallmarks of aging.

## Materials and methods

### Worm strains and maintenance

All *C. elegans*, *C. briggsae*, *C. remanei,* and *C. brenneri* WT and mutant strains used in this study are listed in *Supplementary file 1*. All strains were maintained on Nematode Growth Media (NGM) plates with 50 µg/mL streptomycin (Gibco) and a lawn of OP50-1 bacteria (a gift from M.-W. Tan) from stationary phase cultures. Nematodes were grown at 20℃, with the exception of temperature-

sensitive mutants (*glp-1[e2144], fem-1[hc17], fem-3[q20],* and *spe-9[hc88]*), which were maintained at 15°C (permissive temperature). When temperature sensitive mutants were used for assays, they were grown at the restrictive temperature (25°C). The genotype of strains was verified by genotyping PCR and Sanger sequencing and the strains were backcrossed three times into our laboratory's N2 strain (in addition to the backcrossing that was performed when the mutants were initially isolated).

Because some *C. elegans* strains have been recently reported to inadvertently contain a long-lived allele of the *fln-2* gene (personal communication, Y. Zhao, H. Wang, R.J. Poole and D. Gems [the Worm Breeder's Gazette, 2018]), we verified the *fln-2* genotype using genotyping by PCR followed by Sanger sequencing using the following primers (5'-GGGTGAAGAATGAGAAACACGC and 5'-ATGATGCAGTTTTGCCAACGG). The forward primer (listed first) was used for sequencing. We confirmed that the key strains used in this study—N2 (WT), BA17 (*fem-1[hc17]*), CB4108 (*fog-2[q71]*), CF1903 (*glp-1[e2144]*), JK816 (*fem-3[q20]*), DG1604 (*fog-2[q71]; ceh-18[mg57]*), and GR1034 (*ceh-18 [mg57]*)—did not contain this long-lived allele of *fln-2* and were wild-type for this allele.

## Lifespan assays

*C. elegans, C. brenneri, C. remanei,* and *C. briggsae* lifespan assays were performed in the same manner. Hermaphrodites or females for these assays were age-synchronized with a short (3–4 hr) egg-lay using young (day 3–5 of life), well-fed adult parents. For the female species (*C. brenneri* and *C. remanei*), age-synchronized, virgin females were identified and isolated from males by placing them on fresh plates at the L4 stage. All worms were grown on NGM plates with streptomycin (50 µg/mL) and seeded with OP50-1 bacteria unless RNAi knock-down was performed. In the case of RNAi knock-down, worms were cultured on NGM containing ampicillin (100 µg/mL, Sigma) and IPTG (0.4 mM, Invitrogen). During development, worms were fed HT115 bacteria (grown to stationary phase, RNAi expression induced for 2–4 hr with 0.4 mM IPTG, and the bacteria concentrated to 20x) carrying empty vector (EV). Upon adulthood (day 3 of life), HT115 bacteria (grown to stationary phase, RNAi expression induced for 2–4 hr with 0.4 mM IPTG, and the bacteria concentrated to 20x) carrying the appropriate RNAi clone from the Ahringer RNAi library (*Kamath et al., 2003*) (a gift from A. Fire) or RNAi clones created in this study (see '*C. briggsae* RNAi knock-down'). The inserts of the plasmids encoding the RNAi clones used in this study were sequenced to verify their identity.

For each assay, worms were scored as dead or alive and transferred to new plates daily during the reproductive period and then every other day. Worms were scored as dead if they did not respond to gentle, repeated prodding with a wire pick (90% Pt, 10% Ir) along different points of their body. Worms were scored as censored if they crawled off the media or died due to bagging (internal hatching) or vulval rupture. Data from these censored worms were included up until the point of censorship (see *Supplementary file 2* for all data).

For conditions in which the effect of sexual interactions was assessed, we used one of three methods, as indicated. For the long-term exposure method (described in *Maures et al., 2014*; *Shi and Murphy, 2014*), young males (day 1 to 2 of adulthood) were added to the hermaphrodites at the onset of adulthood. For lifespan experiments in which the hermaphrodites were exposed to males for their entire adulthood (*Maures et al., 2014*), males were added in a 1:1 ratio with hermaphrodites and the number of males remained fixed, even as hermaphrodites began to die or censored. Male worms were replaced every other day at the time the hermaphrodites were transferred to new plates. Male stocks were set up every day for the entirety of the lifespan assay. For the lifespan experiments in which hermaphrodites were exposed to males for only one day (*Shi and Murphy, 2014*), young males were added in a 2:1 male:hermaphrodite ratio. Following 24 hr of exposure, hermaphrodites were moved to new plates and did not encounter a male again throughout their lifespan. For the newly established, short mating-induced demise lifespan experiments, the males and hermaphrodites were only allowed to interact for 2 hr at the age specified for each assay. Males were twice as abundant as hermaphrodites during the mating period. Following a mating period of two hours, mating was assessed by the presence of fluorescent male sperm (see below). These mated hermaphrodites did not interact with an adult male again and worms on plates in which male progeny reached adulthood and could have mated with the hermaphrodites were censored.

Synchronized individuals (hermaphrodites, feminized individuals etc.) were randomly assigned to the 'no males' or '+males' conditions by picking them onto fresh plates in an alternating manner to avoid selection bias. Similarly, the males used for mating with individuals of different genotypes or ages were from the same sets of males and were allocated randomly in an alternating manner. For

each single biological replicate, approximately 35 individuals were placed on each of 2–4 plates (each plate represents a technical replicate). The number of individuals per plate and number of technical replicates were chosen based on field standards (*Lucanic et al., 2017*).

For feminized and sterile mutants, slight modifications were made to the methods. The sterile *glp-1(e2144)* mutant, fertilization defective *spe-9(hc88)* mutant, masculinized *fem-3(q20)* mutant, and WT control parents were used for an egg lay at the permissive temperature (15℃) and following the egg lay, the individuals used for the assay were kept at 25℃ for the remainder of the assay. The feminized *fem-1(hc17)* mutants and the WT control parents were kept at 15℃ for the egg lay and these eggs developed until day 3 of life (adulthood) at 25℃ (the restrictive temperature). Then, both the *fem-1(hc17)* feminized worms and the WT worms were moved to 20℃ for the remainder of the assay. The lower temperature did not impact the feminized phenotype. For the auxin-inducible degradation of the transcription factor SPE-44, the worms were grown on NGM with 1 mM auxin (3-indoleacetic acid, Sigma-Aldrich) from egg lay to adult day one and were then cultured on NGM without auxin for the remainder of their lifespan. These individuals were fully self-sterile as expected (*Kasimatis et al., 2018*). Finally, the *C. elegans fog-2(q71)* and *C. briggsae she-1(v35)* feminized mutants were kept at 20℃ for the entire assay. For these assays, virgin feminized individuals were isolated from males at the L4 stage and were either kept away from males for the entire lifespan or only interacted with males for 2 hr as described.

Lifespan data were plotted as Kaplan-Meier survival curves in Prism 7 and statistical analyses performed using the logrank (Mantel-Cox) test. The number of animals (n) used for each assay and the number of independent biological replicates (N) can be found in *Supplementary file 2*.

## Fluorescent sperm tracking to identify mated hermaphrodites and females for mating efficiency assays and for lifespan assays

To identify and isolate hermaphrodites or females that have mated during a brief period of interacting with males, we modified a previously developed technique (*Stanfield and Villeneuve, 2006*). The day before the mating assay, adult day one males (for *C. elegans*, either *him-5[e1467]* or *nIs128 [Ppkd-2::GFP]; him-8[e1469]*, for *C. briggsae*, *Cbr-him-8[v186]*, and WT for *C. brenneri* and *C. remanei*) were fluorescently labeled by culturing them overnight on NGM plates seeded with 100 µL stationary phase OP50-1 and 5 ng/µL MitoTracker Red CMXRos (Thermo Fisher cat# M7512, resuspended in DMSO [Fisher] at 100x and kept at −20℃ in aliquots) at a density of approximately 100 males per 6 cm plate. On the day of the mating assay, 40 fluorescently-labeled males and 20 unlabeled hermaphrodites were placed on 6 cm NGM plates seeded with OP50-1 bacteria. Animals were allowed to interact with each other and mate for 2 hr at their normal culturing conditions. Following this period, hermaphrodites that received male sperm as a result of mating were identified based on the presence of fluorescence (male sperm) in their uterus and/or spermatheca (*Figure 1E* and *Figure 1—figure supplement 1B–D*) using a fluorescent dissecting microscope or the COPAS large particle biosorter as indicated. In a small number of cases, a hermaphrodite consumed some of the MitoTracker Red CMXRos labeled bacteria resulting in red fluorescence in the gut or throughout the body. When this occurred, the individual was censored from the experiment because of the difficulty in determining if they received male sperm or not.

For mating efficiency assays, 5–7 mating plates (n) were typically used per experimental condition and one to two independent, biological replicates (N) were performed. The number of mated hermaphrodites was compared to the total number of hermaphrodites per plate (mating efficiency = # hermaphrodites with male sperm/total # hermaphrodites).

To better understand the dynamics of mating during a two-hour period (*Figure 2—figure supplement 1E*), we performed a separate assay using a slightly modified version of the mating efficiency method such that two male genotypes were used. Specifically, on each plate 20 young, unlabeled hermaphrodites or feminized individuals interacted with 20 unlabeled *nIs128(Ppkd-2::GFP); him-8 (e1469)* males and 20 MitoTracker Red CMXRos labeled *him-5(e1467)* males for two hours. Following this 2 hr interaction, hermaphrodites that received male sperm from the fluorescently labeled *him-5 (e1467)* males were isolated and placed individually on 3 cm NGM plates to lay eggs. To determine if any of these *him-5(e1467)* mated individuals also mated with a *nIs128(Ppkd-2::GFP); him-8(e1469)* male, we looked for male progeny that carried *nIs128[Ppkd-2::GFP]* transgene. These individuals were thus mated at least twice and received male sperm from at least two males.

For lifespan assays, approximately eight mating plates were set up per condition. Hermaphrodites or females that received male sperm were moved to fresh plates without males at a density of approximately 35 individuals per plate and tracked for the remainder of their lifespan.

## RNA-seq

To better understand the resistance of young hermaphrodites to mating-induced demise, we characterized the transcriptomes of the young, WT hermaphrodites that have a normal lifespan after a brief interaction with males to those that have a shortened lifespan after a brief interaction with males (the self-sperm depleted fem-1[hc17] feminized individuals and the middle-aged, WT hermaphrodites).

WT and fem-1(hc17) mutant individuals were age-synchronized using a brief, 3–4 hr egg lay (see 'Lifespan Assays' above) on NGM plates seeded with OP50-1 bacteria and grown at 25°C during development and the young adult stage. On day 3 of life (egg lay is day 0), the worms were all moved to 20°C for the remainder of their lifespan. This set up resulted in fully penetrant feminized germline phenotypes for the fem-1(hc17) mutants and delayed the age-associated appearance of red autofluorescence in the gut that would have made accurate detection of MitoTracker Red CMXRos labeled male sperm in the uterus and spermatheca difficult. For each of the four biological replicates, egg lays were performed on two separate days such that there was a cohort of age-synchronized young (day 3 of life) and middle-aged (day 7 of life) hermaphrodites of both genotypes on the day of mating. Following the Materials and methods section 'Fluorescent Sperm Tracking to Identify Mated Hermaphrodites for Mating Efficiency Assays and for Lifespan Assays', we identified and isolated young and middle-aged WT and fem-1(hc17) individuals that received male sperm from fluorescently labeled him-5(e1467) males and transferred them to fresh plates. At the same time, unmated hermaphrodites that never encountered a male were also moved to fresh plates at the same density (approximately 30 worms per plate). After approximately 24 hr, 40 worms from each condition (individuals of each age and genotype that either received male sperm or never interacted with males) were picked onto an unseeded NGM plate. These worms were immediately washed twice with ice cold M9 buffer (22 mM $KH_2PO_4$, 42 mM $Na_2PO_4$, 86 mM NaCl, and 1 mM $MgSO_4$) and the worm pellets were flash frozen in liquid nitrogen. The remaining worms were used to measure their lifespan (Figure 2A–D and Supplementary file 2) as described in the section 'Lifespan Assays'.

RNA was extracted from the flash frozen worm pellets with 500 µL Trizol and 200 µL chloroform followed by 250 µL phenol and 200 µL chloroform extractions and finally, an isopropanol precipitation. Remaining DNA was degraded with DNaseI (Promega) and the RNA cleaned with a sodium acetate and ethanol precipitation. RNA quality was measured using Nanodrop spectrophotometry and the Agilent BioAnalyzer Total RNA Nano chip and kit. mRNA enriched cDNA was prepared using 10 ng of total RNA (quantified by Nanodrop spectrophotometry) and the Takara SMART-seq v4 Ultra Low Input RNA kit, with 8 rounds of amplification. Paired-end libraries were made using the Nextera XT DNA library prep kit (Illumina) with 1 ng of cDNA (quantified using the Qubit dsDNA High Sensitivity reagents, Invitrogen) and barcoded using the Nextera XT Index Kit v2 (Illumina). Libraries were purified with 30 µL AMPure XP beads (Beckman Coulter) as directed in the Nextera XT kit. Library quality and quantity were assessed using the Agilent Bioanalyxer High Sensitivity DNA Assay. All samples and biological replicates (A-D) were pooled and sequenced on a single Illumina NextSeq run. Paired-end, 75 base pair sequencing was performed.

## RNA-seq analysis

RNA-seq reads were aligned to the WBcel235 genome and gene read counts were calculated using STAR (version 2.5.4a). Low-coverage genes that had less than one read count per million mapped reads in less than three samples were filtered out. Data were normalized with a variance-stabilizing transformation (DESeq2 version 1.10.1) prior to Principal Component Analysis (PCA) in R (version 3.2.4 and Biobase version 2.30.0). PCA was carried out using the R method (prcomp). Differential expression was calculated using DESeq2 (version 1.10.1). The results from DESeq2 can be found in Figure 2—source data 1. Gene Ontology (GO) enrichment was performed in R using the Fischer's t-test and Benjamini-Hochberg corrected. Full GO enrichment results can be found in Figure 2—

*source data 2*. Heatmaps were generated in R using normalized read counts (variance-stabilizing transformation). Code is available online (https://github.com/brunetlab/Booth_etal_2019.git).

## CEH-18 network analysis

The CEH-18 network was determined using the GeneMANIA webserver (*Warde-Farley et al., 2010*). The default settings were used with the exception of the types of interactions allowed (predicted interactions and protein domain similarity were deselected). Networks were re-drawn based on the GeneMANIA output using Adobe Illustrator to incorporate the DESeq2 output from the RNA-seq (*Figure 2—source data 1*). P-values for enrichment of differentially expressed genes in the network were calculated using the hypergeometric distribution.

## CEH-18 DNA binding motif and ChIP-seq peak analysis

The PSWM file for the CEH-18 DNA binding motif (*Narasimhan et al., 2015*) was downloaded from http://cisbp.ccbr.utoronto.ca and a publicly available CEH-18 ChIP-seq dataset (*Kudron et al., 2018*) was downloaded from http://epic.gs.washington.edu/modERN/. The ChIP-seq peaks were then converted to FASTA format with bedtools 'getfasta' (*Quinlan and Hall, 2010*) to be made compatible with MEME. The MEME Suite AME (with default parameters 'Average odds score' and 'Fisher's exact test' using shuffled input sequences as the control) was used with the CEH-18 motif to determine statistical enrichment of the CEH-18 motif within the CEH-18 binding peaks (*McLeay and Bailey, 2010*). This analysis revealed significant enrichment for the CEH-18 DNA binding motif in the CEH-18 ChIP-seq peaks (p=$4.77\times10^{-5}$), though the enrichment was not strong. As a comparison, the same analysis was performed using the DAF-16 PSWM file (http://cisbp.ccbr.utoronto.ca; *Weirauch et al., 2014*) to test for enrichment within DAF-16 ChIP-seq peaks (http://data.modencode.org; *Contrino et al., 2012*). There was very high enrichment of the DAF-16 DNA binding motif within DAF-16 ChIP-seq peaks (p=$1.44\times10^{-359}$).

We next tested for the enrichment of the CEH-18 DNA binding motif in the sequences surrounding the transcription start sites of the differentially expressed genes from the RNA-seq. The ChIP-Seeker (*Yu et al., 2015*) function 'getPromoters' was used with parameters 'TxDb = TxDb.Celegans.UCSC.ce11.refGene, upstream = 300, downstream = 300' to assign promoters to genes from lists of differentially expressed genes generated from the RNA-seq data with a statistical significance threshold of FDR < 0.05. The resulting bed files were converted to FASTA format and inspected for motif enrichment as above using the MEME suite AME (*McLeay and Bailey, 2010*; *Quinlan and Hall, 2010*).

To identify differentially expressed genes that have a nearby CEH-18 binding site, the publicly available CEH-18 ChIP-seq peaks (*Kudron et al., 2018*) were first de-duplicated to remove any non-unique lines, and the bed file was converted to a gRanges object using readBed() from the 'genomation' package (*Akalin et al., 2015*). The gRanges object was then annotated using annotatePeak() from the 'ChIPSeeker' package (*Yu et al., 2015*) to associate peaks with a proximal gene. The CEH-18 ChIP-seq peaks that were associated with genes differentially regulated in the RNA-seq datasets were then outputted based on the annotated gene associations (from ChIPSeeker; *Yu et al., 2015*). This output is *Figure 4—source data 1*. Using the hypergeometric distribution test, we found that the differentially expressed genes from our RNA-seq (comparing young hermaphrodites to young feminized individuals, mated and unmated) were not significantly enriched for the CEH-18 binding sites. We note that our RNA-seq was performed using whole worms and that the differentially expressed genes from these data are the result of transcripts from all the tissues, many of which may not express *ceh-18*. The lack of transcription factor binding enrichment, particularly in transcription factors acting in a cell- or tissue-specific manner, has been described in *C. elegans* (*Cao et al., 2017*; *Narasimhan et al., 2015*) and is thought to be largely due to cell heterogeneity. Cell- or tissue-specific methods including single-cell RNA-seq could better elucidate the CEH-18 regulatory network in the somatic gonad.

We then generated a list of differentially expressed genes between young hermaphrodites and young feminized individuals (unmated and that mated with males) that have a neighboring CEH-18 binding site by associating ChIP-seq binding peaks with nearest transcription start site using ChIP-Seeker (*Yu et al., 2015*). These data are in *Figure 4—source data 1*. This subset of genes was then used for GO enrichment in R using the Fischer's t-test and Benjamini-Hochberg correction. The

output for the GO enrichment is *Figure 4—source data 2*. As a negative control, we also performed this analysis with an unrelated transcription factor, MAB-5 and these data are presented in *Figure 4—source data 1* and *2*. All code for these analyses are available online (https://github.com/brunetlab/Booth_etal_2019.git).

## *C. briggsae* cross

To create a feminized *C. briggsae* worm that is competent for RNAi knock-down by ingested dsRNA, we crossed the *she-1(v35)* feminized strain to a transgenic strain (*mfIs42[Cel-sid-2; Cel-myo-2::DsRed]*) that is susceptible to ingested dsRNA (*Nuez and Félix, 2012*). Worms that were homozygous for the *she-1(v35)* mutation were determined by PCR followed by Sanger sequencing using the following primers:

5'- CAATTGTCATGCGACCAGATTT −3'
5'- GCTTGTCCGAAACCAATGAAC −3'

The homozygous presence of the *mfIs42[Cel-sid-2; Cel-myo-2::DsRed]* integration was determined by observing transmission of the DsRed marker in all progeny for several generations.

## *C. briggsae* RNAi knock-down

Portions of the highly conserved *Cbr-ceh-18* and *Cbr-vab-1* genes (*Figure 5—figure supplement 2A*) were amplified from *C. briggsae* (strain AF16) genomic DNA and the DsRed transgenic marker (from JU1018 genomic DNA) using Platinum HiFi Supermix (Invitrogen) and the following primers:

*Cbr-ceh-18:* 5'-GGTCCTCGAGGTATTCACCAACGGCAACAAC-3' and 5'-GCGTACTAGTGGTCCTCTTCCTTCTTCTCTTG-3'

*Cbr-vab-1:* 5'-GGTCCTCGAGAGTGTGGATCCGTTGTGATG-3' and 5'-GCGTACTAGTGGAAATCCAACTCACCCTATGA-3'

*dsRed:* 5'-GGTCCTCGAGGAACGTCATCACCGAGTTCAT-3' and 5'-GCGTACTAGTGATGGTGTAGTCCTCGTTGTG-3'

The PCR products and the L4440 vector were digested with SpeI and XhoI (New England Biolabs). Ligation was performed with T4 ligase (New England Biolabs). The ligation products were first transformed into TOP10 chemically competent cells (Invitrogen). After verification of the plasmids by Sanger sequencing, the correct plasmids were transformed into chemically competent HT115 *E. coli*. HT115 bacteria containing these plasmids were used for RNAi knock-down by feeding in *C. briggsae* (*mfIs42[Cel-sid-2; Cel-myo-2::DsRed]*) (see 'Lifespan Assays' above).

## Microscopy

Worms were prepared for imaging using 1 mM sodium azide and mounted on a 2% agarose pad. The images presented and quantified in *Figure 5—figure supplement 3* were taken using a Zeiss Axioskop 2 Plus. All images for an experiment were taken using the same exposure length. DsRed fluorescence was quantified in Fiji (*Schindelin et al., 2012*) using the mean gray value of the pharynx bulb.

## Protein alignment and conservation

To measure the conservation of orthologs of the sperm-sensing proteins CEH-18 and VAB-1, the sequences of these proteins were downloaded from www.wormbase.org (WBcel235) and aligned using MUSCLE v3.8 (*Edgar, 2004a*; *Edgar, 2004b*). These protein alignments were input to the JalView visualization tool (*Waterhouse et al., 2009*) to generate *Figure 5—figure supplement 2A*. Protein domains (pfam) were determined by www.wormbase.org.

## Data and materials availability

Data are available in the main text or supplementary materials. RNA-seq reads are available online at NCBI SRA (PRJNA508378) and all code used for RNA-seq analysis is available online (https://github.com/brunetlab/Booth_etal_2019.git; copy archived at https://github.com/elifesciences-publications/Booth_etal_2019; *Booth, 2019*).

## Acknowledgements

We thank Coleen Murphy and Cheng Shi for sharing their unpublished data. We thank Jennifer Garrison, Coleen Murphy, Patrick Phillips, Cheng Shi, and Anne Villeneuve for helpful discussion and feedback. We thank all the members of the Brunet lab, in particular Chi-Kuo Hu, Salah Mahmoudi, Ravi Nath, Katharina Papsdorf, and Param Priya Singh for helpful discussion and reading the manuscript. We thank Matthew Buckley for independently checking code and Katja Hebestreit for help and advice on the analysis of RNA-seq data. The Stanford Functional Genomics Facility performed the sequencing for the RNA-seq libraries. We thank the *Caenorhabditis* Genetics Center, which is funded by NIH grant P40 OD10440, Abby Dernburg, Ron Ellis, H Robert Horvitz, Patrick Phillips, and Man-Wah Tan for providing *C. elegans* strains used in this study and WormBase. This work was supported by NIH DP1 AG044848 and R01 AG054201 (AB), the Helen Hay Whitney Foundation and NIH K99 AG051738 (LNB), and the Genentech Graduate Fellowship (RWY).

## Additional information

### Competing interests

Travis J Maures: is affiliated with Synthego. The author has no financial interests to declare. The other authors declare that no competing interests exist.

### Funding

| Funder | Grant reference number | Author |
|---|---|---|
| National Institutes of Health | DP1 AG044848 | Anne Brunet |
| National Institutes of Health | R01 AG054201 | Anne Brunet |
| Helen Hay Whitney Foundation | Joan Whitney Payson Scholar | Lauren N Booth |
| National Institutes of Health | K99 AG051738 | Lauren N Booth |
| Genentech Foundation | Genentech Foundation Predoctoral Fellow | Robin W Yeo |

The funders had no role in study design, data collection and interpretation, or the decision to submit the work for publication.

### Author contributions

Lauren N Booth, Conceptualization, Software, Formal analysis, Funding acquisition, Validation, Investigation, Visualization, Methodology, Writing—original draft, Writing—review and editing; Travis J Maures, Conceptualization, Methodology, Writing—review and editing; Robin W Yeo, Software, Funding acquisition, Validation, Writing—review and editing; Cindy Tantilert, Formal analysis, Investigation; Anne Brunet, Conceptualization, Supervision, Funding acquisition, Writing—original draft, Project administration, Writing—review and editing

### Author ORCIDs

Lauren N Booth  https://orcid.org/0000-0003-3072-6235
Anne Brunet  https://orcid.org/0000-0002-4608-6845

### Decision letter and Author response

Decision letter https://doi.org/10.7554/eLife.46418.028
Author response https://doi.org/10.7554/eLife.46418.029

## Additional files

### Supplementary files

• Supplementary file 1. Nematode strains used in this study. The complete list of all strains used in this study, with their genotype and source listed.
DOI: https://doi.org/10.7554/eLife.46418.020

• Supplementary file 2. Lifespan assay results. The data for the lifespan assays displayed in the figures as well as lifespan assays whose plots are not shown in the manuscript. Each set of lifespan assays performed together is separated from the other sets of assays by a blank row in the table. 'Temp.' describes the temperature at which the worms were grown. Note that 25→20 indicates that the hermaphrodites and feminized individuals were grown during development at the restrictive temperature and then shifted to a lower temperature on day 3 of life as described in the methods. All *p*-values were determined using Mantel-Cox log-ranking.
DOI: https://doi.org/10.7554/eLife.46418.021

• Supplementary file 3. Mating efficiencies. The mating efficiencies performed. The individuals whose mating efficiencies were measured are underlined in each set of mating partners. Successful mating was determined by the presence of fluorescent male sperm in the spermatheca or uterus. The presence of fluorescent male sperm is indicative of fertilization (see *Figure 1—figure supplement 1B,C*) though this was not specifically measured in this assay. See *Source data 1* for a complete list of the mating efficiencies for each plate of 20 hermaphrodites (or germline mutants) with 40 males that were used to calculate the median and *p*-values.
DOI: https://doi.org/10.7554/eLife.46418.022

• Source data 1. The raw data that comprise *Supplementary file 3*.
DOI: https://doi.org/10.7554/eLife.46418.023

• Transparent reporting form
DOI: https://doi.org/10.7554/eLife.46418.024

### Data availability

Sequencing data have been deposited in NCBI SRA under accession code PRJNA508378.

The following dataset was generated:

| Author(s) | Year | Dataset title | Dataset URL | Database and Identifier |
|---|---|---|---|---|
| Booth LN, Maures TJ, Yeo RW, Brunet A | 2019 | Transcriptomic profiling of C. elegans hermaphrodites and feminized (fem-1) individuals following a 2 hour interaction with males or without a male interaction | http://www.ncbi.nlm.nih.gov/bioproject/?term=PRJNA508378 | NCBI Sequence Read Archive, PRJNA508378 |

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
