## [Decision Letter]

Thank you for submitting your article "Self-sperm induce resistance to the detrimental effects of sexual encounters with males in hermaphroditic nematodes" for consideration by *eLife*. Your article has been reviewed by three peer reviewers, including Matt Kaeberlein as the Reviewing Editor and Reviewer #1, and the evaluation has been overseen by Patricia Wittkopp as the Senior Editor. The following individual involved in review of your submission has also agreed to reveal their identity: George Sutphin (Reviewer #2).

The reviewers have discussed the reviews with one another and the Reviewing Editor has drafted this decision to help you prepare a revised submission.

All of the reviewers are in agreement that the work described here is of general interest and generally high quality. A few relatively minor concerns were raised that the authors are encouraged to address in a revised submission.

Essential revisions:

1) While the reviewers felt that the data were quite interesting, there were some concerns as to whether the interpretation was sufficiently developed in the context of natural environments and evolutionary conservation. The authors are encouraged to develop this in the Discussion.

One reviewer commented: "In the Introduction, the authors state that mating in mammals can have negative impacts on health. What would such negative impacts be and could the protective mechanisms for male-induced demise by self-sperm be also conserved in mammals? This is especially important to address since the authors make strong claims that their results could provide "new insights into the role of gametes in 'resetting' the aging clock in all species." Really, this statement is overstated and is reiterated three times, i.e., in Introduction, Results and Discussion, but should be a discussion point only without any direct data as support. To this end, the authors may want to consider modulation of the sperm-sensing pathway in *C. briggsae*. Moreover, would overexpression of components of the sperm-sensing pathway including in older animals be beneficial?"

Another reviewer commented: "The authors of both manuscripts make several assumptions and statements about how and why these mechanisms evolved, but it seems to me that a careful consideration of what these animals experience in the wild may be lacking. Is it ever the case that wild, reproductively mature nematodes are likely to experience a high enough density of males for some of these mechanisms to be engaged? Are long and frequent sexual encounters with males something that ever happens or is this an unnatural state that can only be induced in a laboratory? While it is plausible that the protection conferred in young hermaphrodites from brief sexual encounters might be important in wild populations, it seems unlikely (to me at least) that sperm-depleted adult hermaphrodites will be exposed to a high density of males for a long enough time that the lifespan shortening effects would be relevant. In fact, is relatively modest lifespan shortening under laboratory conditions even something that would ever matter in the real world? Is laboratory adult lifespan something that matters at all in natural populations? I realize this is largely speculation since we don't know much about aging of natural populations of nematodes, but as far as I know, there's no evidence for very old worms in nature.

While I don't think this reduces the interest and importance of this work, I do think it's important to consider the natural environment when trying to understand how and why these biological interactions came about. Is a slight shortening of adult lifespan really detrimental? One could argue that this isn't clear, since most worms in nature aren't (probably) going to live long enough to experience this effect. Some consideration and discussion of the likely relevance of these mechanisms on fitness in the real world would be valuable."

2) There were shared concerns that some experiments were not sufficiently replicated, or if triplicate replication was performed it was not apparent in some cases. In particular, the authors should assure that they have three biological repeats for their RNAi experiments. It also appears as if the COPAS experiment was only done once.

3) The RNASeq is not well integrated in the overall study. The authors could at least discuss a possible connection between their identified transcription factor and the target genes in their RNASeq. As it is, this part really 'dead ends' in the manuscript.

---

## [Author Response]

Essential revisions:1) While the reviewers felt that the data were quite interesting, there were some concerns as to whether the interpretation was sufficiently developed in the context of natural environments and evolutionary conservation. The authors are encouraged to develop this in the Discussion.

We thank the reviewers for this suggestion. We have now included additional discussion of the natural environment in which nematodes evolved and the evolutionary conservation of these phenomena (subsection “Young hermaphrodites are protected from demise induced by a brief mating with males” and Discussion).

One reviewer commented: "In the Introduction, the authors state that mating in mammals can have negative impacts on health. What would such negative impacts be and could the protective mechanisms for male-induced demise by self-sperm be also conserved in mammals? This is especially important to address since the authors make strong claims that their results could provide "new insights into the role of gametes in 'resetting' the aging clock in all species." Really, this statement is overstated and is reiterated three times, i.e., in Introduction, Results and Discussion, but should be a discussion point only without any direct data as support. To this end, the authors may want to consider modulation of the sperm-sensing pathway in C. briggsae. Moreover, would overexpression of components of the sperm-sensing pathway including in older animals be beneficial?"

We agree with the reviewer. We have now included two examples of the effect of sexual interactions on mammalian health in the Introduction. Whether the mechanisms of male-induced demise or the downstream pathways engaged by self-sperm to protect against the deleterious effects of mating are conserved outside of nematodes is unknown. To better reflect this state of knowledge and as suggested by this reviewer, we have removed the statements regarding the possible roles of gametes in aging from the Results and Introduction sections of the revised manuscript.

We also thank the reviewer for their interesting suggestion to manipulate the sperm-sensing pathway in *C. briggsae* to test if this is beneficial, notably in older animals. We have now included new experiments to test the role of the *C. briggsae* sperm-sensing pathway orthologs in the protection from mating-induced demise. We find that feeding *Cbr-ceh-18* and *Cbr-vab-1* RNAi bacteria to feminized, transgenic *C. briggsae* that are competent for RNAi by feeding (Nuez and Felix, 2012) resulted in protection of middle-aged hermaphrodites following mating with a male (new Figure 5E, F). We also verified that *Cbr-ceh-18* and *Cbr-vab-1* RNAi were effective as they lead to increased number of unfertilized oocytes in the uterus and a reduction in the compaction of the oocytes in the gonads (new Figure 5—figure supplement 3), similar to *C. elegans* (Miller et al., 2003). Thus, the role of the sperm sensing pathway appears to be conserved in *C. briggsae* and has evolved to mediate resistance to mating-induced demise at least twice in hermaphroditic nematodes. These experiments are described in the third paragraph of the subsection “The ability of self-sperm to promote resistance to mating with males evolved independently twice in nematodes” and have been included as new Figure 5E, F, Figure 5—figure supplement 3. All three replicates of the lifespan assays are in new Supplementary file 2.

Another reviewer commented: "The authors of both manuscripts make several assumptions and statements about how and why these mechanisms evolved, but it seems to me that a careful consideration of what these animals experience in the wild may be lacking. Is it ever the case that wild, reproductively mature nematodes are likely to experience a high enough density of males for some of these mechanisms to be engaged? Are long and frequent sexual encounters with males something that ever happens or is this an unnatural state that can only be induced in a laboratory? While it is plausible that the protection conferred in young hermaphrodites from brief sexual encounters might be important in wild populations, it seems unlikely (to me at least) that sperm-depleted adult hermaphrodites will be exposed to a high density of males for a long enough time that the lifespan shortening effects would be relevant. In fact, is relatively modest lifespan shortening under laboratory conditions even something that would ever matter in the real world? Is laboratory adult lifespan something that matters at all in natural populations? I realize this is largely speculation since we don't know much about aging of natural populations of nematodes, but as far as I know, there's no evidence for very old worms in nature.While I don't think this reduces the interest and importance of this work, I do think it's important to consider the natural environment when trying to understand how and why these biological interactions came about. Is a slight shortening of adult lifespan really detrimental? One could argue that this isn't clear, since most worms in nature aren't (probably) going to live long enough to experience this effect. Some consideration and discussion of the likely relevance of these mechanisms on fitness in the real world would be valuable."

We thank the reviewer for the thoughtful comments regarding our interpretations of the potential evolutionary significance of our findings. As suggested by the reviewer, we have now included a discussion of the relevance of this phenomenon in the wild. We now discuss studies of wild populations of *C. elegans* that indicate that males and hermaphrodites interact in the wild, though these interactions are rare. We agree with the reviewer that long interactions with males are unlikely to be frequent in the wild and that a brief interaction, as we have more closely modeled here, is more plausibly relevant to the natural state. We now also include a discussion of whether lifespan shortening in the wild would be (dis)advantageous to males and hermaphrodites. These revisions can be found in the Discussion.

2) There were shared concerns that some experiments were not sufficiently replicated, or if triplicate replication was performed it was not apparent in some cases. In particular, the authors should assure that they have three biological repeats for their RNAi experiments. It also appears as if the COPAS experiment was only done once.

We agree that this is very important. In the revised version of the manuscript, we have included additional replicates for both the COPAS large particle biosorter experiment and the sperm-sensing RNAi experiments. Some of these additional replicates were performed by an independent investigator (Cindy Tantilert), who is included as an author of the revised manuscript. These replicates have been included in Supplementary file 2 and the replicates that were performed by an independent investigator are indicated.

3) The RNASeq is not well integrated in the overall study. The authors could at least discuss a possible connection between their identified transcription factor and the target genes in their RNASeq. As it is, this part really 'dead ends' in the manuscript.

We agree this is an important point, and during the revision we have further analyzed our RNA-seq data to test the relationship between the target genes and the identified transcription factor CEH-18. Specifically:

- We have now performed a network analysis of CEH-18 by using the GeneMANIA web browser. We found a significant enrichment of differentially expressed genes (young hermaphrodites versus young feminized individuals) in this CEH-18 network. These results are presented in new Figure 4F and new Figure 4—figure supplement 1A, B.

- We performed CEH-18 motif and ChIP-seq binding analysis using publicly available data (Kudron et al., 2018; Narasimhan et al., 2015). A subset of genes differentially expressed between young hermaphrodites and feminized individuals had CEH-18 binding sites in their regulatory regions, although there was no significant enrichment (perhaps because heterogenous cell populations in *C. elegans* mask the tissue-type specific binding or gene expression (Cao et al., 2017; Narasimhan et al., 2015). We have added these results in main text, new Figure 4—source data 1, and Materials and methods (subsections “Self-sperm protect hermaphrodites by triggering a sperm-sensing pathway that normally affects the germline to protect the soma”, second paragraph and “CEH-18 DNA binding motif and ChIP-seq peak analysis”).

- We have performed GO enrichment analysis on the subset of genes regulated by the presence of self-sperm that contain CEH-18 binding sites. We find that these genes are enriched for several GO-terms involved in the regulation of lifespan, and could mediate the protective effects of self-sperm. We have included these results in new Figure 4G, new Figure 4—figure supplement 1C, and new Figure 4—source data 2.

Together, these analyses suggest that changes in CEH-18 activity may contribute to the gene expression differences that underlie sensitivity and resistance to mating-induced death.